# Reproductive Health Practices in Spanish Women Who Underwent Voluntary Termination of Pregnancy

**DOI:** 10.3390/diseases11010037

**Published:** 2023-02-27

**Authors:** M. V. Lapresa-Alcalde, A. M. Cubo, M. Alonso-Sardón, M. J. Doyague-Sánchez

**Affiliations:** 1Department of Obstetrics and Gynecology, Hospital Virgen de la Concha, 49022 Zamora, Spain; 2Department of Obstetrics and Gynecology, Hospital Universitario de Salamanca, University of Salamanca, Institute for Biomedical Research of Salamanca (IBSAL), 37007 Salamanca, Spain; 3Preventive Medicine and Public Health Area, University of Salamanca, Institute for Biomedical Research of Salamanca (IBSAL), Tropical Disease Research Centre of the University of Salamanca (CIETUS), 37008 Salamanca, Spain

**Keywords:** voluntary termination of pregnancy, contraceptive methods, elective abortion, contraception, termination of pregnancy

## Abstract

**Background:** The incidence of Voluntary Termination of Pregnancy (VTP) is an important indicator of unplanned pregnancies and the differences in the functioning of contraceptive services and the effectiveness of their use. Its analysis is essential for monitoring the well-being of women and their partners. Our aim was to analyse the socio-demographic profile of women who request voluntary termination of pregnancy in the province of Salamanca, as well as their satisfaction with the intervention and its influence on their contraceptive methods. **Methods:** An intervention study (before-after) designed without a control group, including all women requesting a voluntary termination of pregnancy through the Salamanca Public Health System. Socio-demographic and reproductive health variables were used. After the termination of pregnancy, a satisfaction survey and analysis of consequences were carried out. **Results:** A total of 176 surveys were obtained. Women who underwent VTP in Salamanca were between 20 and 25 years old, had secondary education but were still studying or working, lived alone and had no children. The most commonly used contraceptive method was the condom (55%), followed by the pill (25%). The most frequent reason for termination of pregnancy was economic (47.7%). The abortion entailed a significant change in contraception. Whereas before the abortion only 34% used a hormonal method, 66% were willing to use one afterwards (*p* = 0.006). **Conclusion:** Reproductive health education needs to be improved so that couples use reliable contraceptive methods appropriately. Although women are generally satisfied with the care received during abortion, they would prefer better accessibility to the procedure and more comprehensive information about the process from a neutral stance.

## 1. Introduction

Voluntary Termination of Pregnancy (hereinafter referred to as VTP) occurs when a woman decides to end her pregnancy by medical procedures before the full term. It is a global phenomenon that responds to sociodemographic patterns, such as gestational age, the presence or absence of a stable partner, migration, a poor educational level, the characteristics of each country and the religious context [1,2,3]. On the other hand, VTP has negative consequences for women’s health, influences female sexuality and women’s emotional reactions linked to a sense of loss.

In the world there are different legal situations regarding abortion. In Spain, abortion is currently ruled by Organic Law 2/2010, of 3 March, on sexual and reproductive health and the voluntary interruption of pregnancy [4]. During the first 14 weeks of gestation, women over 18 years of age can proceed, freely and without any oversight, to voluntarily terminate pregnancy. They can also proceed up to the 22nd week if there is a serious risk to the life or health of the pregnant woman. A specialized doctor other than the one who will perform the intervention must confirm the diagnosis. Finally, this law does not establish gestational age limit for terminations due to an extremely serious and incurable disease in the foetus.

The incidence of VTP is an important indicator of unplanned pregnancies, the effectiveness of contraceptive services and use of effective contraception [2]. The rate of VTP in Spain has decreased between 2008 and 2016 from 11.78 to 10.36 per 1000 women between 15 and 44 years old, respectively, according to the National Institute of Statistics. After that, it has progressively increased to reach 11.53 VTP per 1000 women in 2019 [3,4]. In 2019, 99,149 terminations were performed in Spain, 81% of which were carried out in private outpatient facilities [3]. Individual socioeconomic characteristics such as non-EU immigrants, low levels of public expenditure on non-university education, average cost of house ownership and average number of children are strongly associated with induced abortion rates in Spain [3].

The aim of this study is to analyse the socioeconomic characteristics and reproductive health practices of women who request a VTP in the province of Salamanca, as well as their satisfaction with the procedures they undergo, in order to improve health care for these patients and reduce the rate of unplanned pregnancies.

## 2. Methods

### 2.1. Study Design and Setting

An intervention study, with before-after assessment, no control group and no randomization, was conducted in women with an unplanned pregnancy who requested a VTP through the public healthcare system in Salamanca. This hospital centralizes all abortion requests through the Social Security of that province. The religious environment is not significant. However, demographic factors have been changing as the population has been aging and the birth rate has been declining. Methodology of this research is characterized as descriptive, we observed the study subjects’ behaviour and recorded qualitative-quantitative data before and after the intervention. To collect these data, we designed pre- and post-intervention surveys: termination of pregnancy by medical or surgical treatment performed, by agreement, in a private clinic in Valladolid, a city 90 km away from Salamanca.

### 2.2. Study Subjects, Inclusion and Exclusion Criteria

Inclusion criteria were all women who accepted and completed at least one of the surveys. Exclusion criteria were refusal to enrol in the study or suffered miscarriage during the VTP process.

In 2018, 175 VTPs were reported in Salamanca (175/52,562; 3.3 per 1000 women aged 15–44 years) and 204 VTPs (204/51,648; 3.9 per 1000 women aged 15–44 years) in 2019. In total, 379 VTPs were reported in Salamanca over the data collection period (2018–2019). During the pilot study (March to August 2018), 48 surveys were collected. During the data collection period (September 2018 to December 2019), 128 pre-VTP and 70 post-VTP surveys were obtained. Finally, a total of 176 women voluntarily participated in the pre-VTP and 70 in the post-VTP survey, accounting for 46.4% of the 379 reported VTP during the study period. The rest declined to participate due to the emotional situation.

### 2.3. Study Instruments

A structured, self-completed pre- and post-survey questionnaire was used to collect primary data. The pre-VTP survey analysed the socio-economic characteristics of the women as well as their contraceptive practices. The subsequent questionnaire allowed us to find out their satisfaction with the intervention (difficulty of the procedures, how the journey and the intervention had affected them) and the effect of this process on their contraceptive practices (such as changing contraceptive methods and proposals for improvement).

The sample set was collected in the family planning clinic. Here, the first questionnaire was completed and a telephone number was requested for the post-abortion satisfaction survey.

### 2.4. Data Analysis

Data collection and analysis of the results were carried out using the Statistical Package for the Social Sciences 25.0 statistical software. A descriptive analysis and a study of the association between variables was carried out. Numbers (n) and percentages (%) were used to describe the basic variables of this research. In the bivariate analysis, a Chi-square (χ^2^) test was used to test differences between two categorical variables and the measured outcome was expressed as the odds ratio (OR) together with the 95% confidence interval (CI) for OR. A *p*-value of *p* < 0.05 was considered statistically significant.

### 2.5. Ethics Statement

This study was approved by the Clinical Research Ethics Committee of the University Hospital of Salamanca (Code: PI 2018 08 103). All data were kept confidential and processed anonymously in accordance with the requirements of Law 3/2018 of 5 December on the Protection of Personal Data and guarantee of digital rights.

## 3. Results

The average age of the patients was 27 years old, with the majority age group being between 20 and 25 years. Most of the women interviewed had completed secondary education/high school training and were either working or studying. More than half of them lived alone and had an income below the minimum wage in 2018 (€736). Most of the women were Spanish, although Colombia was the most frequent foreign nationality with 7 cases. Most of the patients who participated in the study had no previous children (Table 1).

### 3.1. Pre-VTP Questionnaire

Table 2 summarizes data collected in the pre-VTP questionnaire. The most commonly used contraceptive method was the condom, followed by the pill. The main reason for refusal of hormonal methods was fear of side effects. The most frequently used sources of information were the primary care physician and the midwife.

The most frequent age of first sexual intercourse was between 14 and 17 (56.3%), decreasing significantly among the women in our sample. Thus, 73% of women under 25 started sexual activity before the age of 18, while almost half (47.4%) of women over 25 started after the age of 18 (*p* = 0.028). In addition, the proportion of women reporting having received information on sexual health is higher among younger women (<25 years), 89.7% vs. 73.4% [OR = 3.1; 95% CI, 1.3–7.3; *p* = 0.007].

There are significant differences (*p* = 0.028) between the cause of pregnancy and work occupation. Housewives were the most likely (62.5%) to acknowledge not using a contraceptive method as a cause of pregnancy, while more than half (52.2%) of female students attributed it to misuse.

Only 10% of the respondents had taken emergency contraception (EC) after the sexual intercourse that had caused this pregnancy. However, 40% had used it on occasion.

The most frequent reason for abortion was financial, followed by being considered too young to have a child. Seventy per cent of the women interviewed had not had a previous abortion, but this practice was twice as common among women over 25 compared to younger women (31.1% vs. 15.5 [OR = 2.4; 95% CI, 1.1–5.3; *p* = 0.020]). The most common support perceived in this situation was from the partner, although 15 women reported being alone in the process as they had not told anyone. Younger women also received more support (95.3% vs. 85.3% OR = 3.5; 95% CI, 1.1–11.4; *p* = 0.031).

### 3.2. Post-VTP Questionnaire

Table 3 shows data collected in the post-VTP questionnaire. The majority of patients found the procedures to obtain an abortion easy. Women with lower incomes reported significantly less difficulty (*p* = 0.028). With respect to travelling to Valladolid, most of the patients considered that it was not a problem, and in some cases, they even preferred to travel to Valladolid rather than to Salamanca (Table 3). Only in a minority of cases did they have major travel problems. However, improving accessibility by performing the abortion in Salamanca was the most frequently demanded aspect.

The abortion marked a before and after in the woman’s life for 24%, while for the rest, although they remembered it 2–3 months after the procedure, claimed it did not affect their daily life or they had already forgotten about it. At this point, 75% of the women wanted to switch or had already switched to a more effective method of contraception, 59% of them being hormonal (31.3% prior to the abortion). If we analyse the 70 patients who attended the follow-up and completed both questionnaires, and compare their MACs, we see how the groups are inverted and differ significantly (*p* = 0.006) before and after the abortion (Figure 1).

## 4. Discussion

Most VTP are the result of an unplanned conception, but the reasons for the decision to abort are multiple and complex. Although it solves part of the problem of unplanned pregnancy, at the same time it causes other public health problems. It is never harmless, nor is it desirable for women’s health, and it also causes a cost to the individuals involved and to society. The doctor-patient interaction that takes place in VTP clinics does not allow us to know the circumstances associated with unplanned pregnancies and the decision to abort, nor how women perceive the care received. These findings justify the interest of this research. In addition, national studies on socio-demographic issues and contraceptive practices of women requesting VTP are old and date back to the previous Organic Law 9/1985 [5].

The age distribution of this study is consistent with other national [6,7] and international [8] studies. The decision to terminate is related to the time of life of the patient. Completing studies and finding a job are frequent reasons for termination in young women, while fulfilled reproductive desires, family reasons or the employment situation are the reasons for this decision in older women [9].

The educational level of the women in this study does not demonstrate the gradient of social inequalities that do appear in the results of other publications [10,11], where women with no education are more at risk of having an abortion. In our sample, most of the women who terminate their pregnancies have an intermediate level of education. This difference may be due, on the one hand, to an evolutionary change in society, where women who are in a period of training or integration in the labour market want to delay the birth of their first child. On the other hand, knowledge of the existence of the resources available for abortion through the public health system and how to access them requires a certain level of education and culture, which may be the reason for the low proportion of women with no education in our sample [12].

Nevertheless, the economic level seems to be decisive when it comes to termination, as only 32% of the sample had an income above the minimum wage and this was named as one of the first reasons for their abortion. However, it must be taken into account that in public health care (the area where this study was carried out), termination is a free intervention, in contrast with in private medicine. Therefore, finding lower incomes in our sample may be a selection bias.

In line with Spanish studies [6,7,13], the most common contraceptive was the condom (55%), followed by the pill (25%). In our sample, women who reported not using any method accounted for 18%, compared to 36% in Serrano’s study and in the study conducted in Sweden in 2002 [7,14], or 27% among adolescents in Portugal [15]. Although the use of reliable contraceptive methods such as condoms and the pill has increased over the years, their effectiveness depends on adherence and how they are used. In Moreau’s study of 1525 adolescent terminations in France, 84% of condom users identified condom slippage or breakage as the cause of pregnancy [16]. In our sample, the most frequently reported method in combination with other contraceptives was the condom. Although one might think that this is due to the use of the “dual method”, this hope collapses when we see that they have marked it together with pull-out or with “no method”. This shows that condoms are not being used properly, and the proof of this is these unplanned pregnancies that end in termination. As in the literature [7,14], in our study there was a low rate of emergency contraception use in the current pregnancy, although these women are frequent users of EC. This demonstrates on the one hand a low awareness of the risk after intercourse that led to pregnancy, and on the other hand, some use of EC as a regular contraceptive method.

Although education on sexuality and reproductive health is increasing in the academic environment of young people [17], as can also be observed in our sample, more effort is still needed to ensure that, when the population needs more information on contraception, they turn to reliable sources such as health professionals. The family planning consultation should be easily accessible to the whole population, creating an environment of trust and, as far as possible, de-medicalised [18].

In addition, sexual behaviours are changing in adolescence, with an increase in the early onset of intercourse and the number of sexual partners [19,20]. In the 2018 survey of the Spanish Society of Contraception, there is a downward trend in the age of sexual debut, with an average age of 18.13 years [21]. In line with national data, in our study, women younger than 25 years start intercourse significantly earlier than older women [22]. These behaviours have short and long-term consequences and it is therefore vital to establish adequate reproductive health education before these risky behaviours take place.

Although most of the women in our sample did not have a previous termination, this variable may be under-represented in the public health system. In a study carried out in Seville in 2005 with women who were preparing to have an abortion, those who were older and had more previous terminations were concentrated in the private system [23].

The evidence available to us has shown that abortion does not increase the risk of mental illness compared to carrying a pregnancy to term [24,25,26]. However, abortion affects them psychologically, and this may cause them to decide in the first few months to use a more reliable method of contraception. They may even agree to use hormonal contraception that they had previously rejected. Face-to-face or telephone follow-up after termination in the family planning clinic could improve contraceptive counselling and adherence to effective methods of contraception, not only at the time after the abortion, but in the longer term [27].

### 4.1. Strengths and Limitations

The sample collected during 2018 and 2019 did not reach the representative sample size. However, it should be borne in mind that participation in the study was voluntary and the emotional situation in which the women found themselves when they attended the dating ultrasound meant that many of them were unable to concentrate on carrying out a survey and declined participation.

As a study based in the public health system, the results may be altered by selection bias. Surveys in both the public and private spheres would provide a more complete picture of the epidemiological profile and reproductive practices of women who decide to terminate a pregnancy.

To the best of our knowledge, this is the first Spanish study that collects the epidemiological and reproductive health profile prior to abortion and follows up to analyse changes in the contraceptive practices of these women, as well as their personal assessment of abortion. Most of the national studies that analyse abortion are cross-sectional studies based on the registers of the Autonomous Communities and respond to the previous Organic Law 9/1985 [5,10,12]. On the other hand, we have not been able to find in the literature any publication on the assessment of the interprovincial displacement that women in Communities such as Castilla y León are obligated to undergo in order to obtain a termination through the public health system.

### 4.2. Clinical Implication

The purpose of this research was to update knowledge on factors related to VTP in Spain in order to implement more effective sexual and reproductive health promotion strategies in our Health Area. All women should have access to adequate information and the possibility to plan their reproductive life with quality, confidentiality and safety. In view of the complexity of the elements affecting VTPs, it is necessary to continue research to better understand the conditions under which they occur, to understand their implications for safety and quality of care, and to minimize potential health problems for women, both physical and mental.

## 5. Conclusions

In order to reduce unplanned pregnancies, and therefore the number of abortion complications, it is necessary to improve training in reproductive health so that couples use reliable contraceptive methods in an appropriate way, and to promote the use of family planning consultations to solve contraceptive needs. Although women are generally satisfied with the care received during the abortion, they would rather have better accessibility to the procedure in their own city and more comprehensive information about the process from a neutral stance.

## Figures and Tables

**Figure 1 diseases-11-00037-f001:**
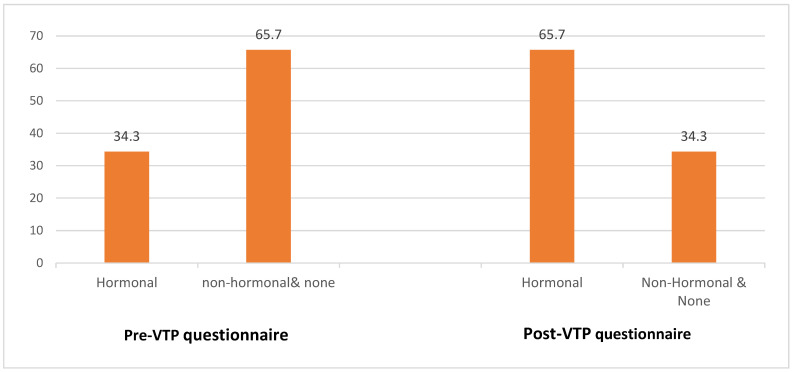
Contraceptive methods: Percentage distribution before and after Voluntary Interruption of Pregnancy.

**Table 1 diseases-11-00037-t001:** Socio-demographic variables collected in the study.

Variable	Categories	TOTAL (176)
		Frec. (%)
Age	15–19 years	28 (15′9)
	20–25 years	63 (35′7)
	26–30 years	39 (22′2)
	>30 years	45 (25′6)
Missing		1 (0′6)
Educational level	Primary	28 (15′9)
	High school	106 (60′2)
	University	40 (22′8)
Missing		2 (1′1)
Proffesion	Student	54 (30′7)
	Employee	55 (31′3)
	Unemployed	37 (21)
	Housewife	20 (11′4)
	Self-employed	8 (4′5)
Missing		6 (3′4)
Address	Alone	109 (61′9)
	Couple	53 (30′1)
	Parents	7 (4)
Missing		7 (4)
Monthly income	<736€	113 (64′2)
	737–1500€	56 (31′9)
	>1500€	5 (2′8)
Missing		2 (1′1)
Birth country	Spain	142 (80′7)
	Other	32 (18′2)
Missing		2 (1′1)
Children	0	81 (46)
	1	23 (13)
	2	17 (9′7)
	3	6 (3′4)
	5	1 (0′6)
Missing		48 (27′3)

**Table 2 diseases-11-00037-t002:** Overall results of the questionnaire prior to the voluntary termination of pregnancy.

Variable	Categories	TOTAL (176)
CONTRACEPTION		**Frec. (%)**
Contraceptives ^1^	Condom	79 (44′9)
	Pill	37 (21)
	Inj./IUD/Implant	3 (1′7)
	Withdrawal	9 (5′1)
	Ring/patch	3 (1′7)
	None	26 (14′8)
	Condom + withdrawl/none	11 (6′3)
	Condom + pill	7 (3′9)
	Missing	1 (0′6)
Reason for rejecting hormonal methods ^1^	Secondary effects	49 (27′8)
	Difficulty of use	4 (2′3)
	Low pregnancy possibility	2 (1′1)
	Lack of information	18 (10′2)
	Contraindicated	8 (4′5)
	Missing	95 (53′9)
Contraceptive method information ^1^	Friends	16 (9′1)
	Internet	12 (6′8)
	Primary care physician	43 (24′4)
	Midwife	42 (23′9)
	Gynecologist	41 (23′3)
	Other	24 (13′6)
	Friends and internet	10 (5′7)
	Missing	8 (4′5)
If they have no method of contraception ^1^	Withdrawal	65 (36′9)
	Buy condom	52 (29′5)
	Abstention	31 (17′6)
	Emergency contraception	20 (11′4)
	Confident that she will not become pregnant	18 (10′2)
	Missing	6 (3′4)
Alcohol: less condom use	Yes	27 (15′3)
	No	120 (68′2)
	Sometimes	19 (10′8)
	Missing	10 (5′7)
REPRODUCTIVE HEALTH		
First sexual intercourse	10–13 years	3 (1′7)
	14–17 years	100 (56′8)
	>18 years	59 (33′5)
	Missing	14 (8)
Number of sexual partners	Less than 2	42 (23′9)
	2 to 5	90 (51′1)
	More than 5	29 (16′5)
	Missing	15 (8′5)
Sexuality information	Yes	136 (77′3)
	No	31 (17′6)
	Missing	9 (5′1)
	Sufficient	73 (41′5)
	Insufficient	48 (27′3)
	Missing	55 (31′2)
	Family environment	13 (7′3)
	Academic setting	64 (36′2)
	Both	27 (15′2)
	Missing	73 (41′3)
Knowledge of LARCs	No	107 (60′8)
	A little	36 (20′5)
	Yes, with correct/incorrect example	16/3 (10′8)
	Missing	14 (7′9)
	Option for her	45 (25′6)
	Not for her	16 (9′1)
	Missing	115 (65′3)
TERMINATION OF PREGNANCY		
Cause of pregnancy	No method	57 (32′4)
	Misuse	62 (35′2)
	Do not know	41 (23′3)
	Missing	16 (9′1)
EC in this pregnancy	No	144 (81′8)
	Yes, first 24 h	10 (5′7)
	Yes, >24 h	7 (4)
	Missing	15 (8′5)
EC in other occasions	No	91 (51′8)
	Once	49 (27′8)
	More than 1	21 (11′9)
	Missing	15 (8′5)
Cause of Interruption ^1^	Too young	58 (32′9)
	Economy	84 (47′7)
	Disrupts personal development	44 (25)
	Family or partner pressures	14 (7′9)
	Couple instability	48 (27′3)
	More children	10 (5′7)
	Unwilling/afraid	6 (3′4)
	Lack of support	1 (0′6)
	Mistreatment	1 (0′6)
	Advanced age	1 (0′6)
	Risky pregnancy	7 (3′9)
	Missing	18 (10′2)
Previous interruptions	No	123 (69′9)
	Yes	37 (21)
	Missing	16 (9′1)
	1	30 (17)
	2	6 (3′4)
	>2	1 (0′6)
Support ^1^	No	15 (8′5)
	Yes	146 (83)
	Missing	15 (8′5)
	Partner	94 (53′4)
	Family	54 (30′7)
	Friend	58 (32′9)

Inj: injectable. IUD: intrauterine device. LARCs: long-acting reversible contraceptive methods, EC: emergency contraception. ^1^ There were multiple answers.

**Table 3 diseases-11-00037-t003:** Overall results of the post-abortion questionnaire.

Variable	Categories	TOTAL (80)
SATISFACTION WITH THE PROCESS		**Frec. (%)**
Formalities	1 = easy to execute	35 (43′7)
	2	18 (22′5)
	3	11 (13′8)
	4	10 (12′5)
	5 = too many	6 (7′5)
	Missing	_
Travelling	I prefer it this way	13 (16′3)
	Not relevant	22 (27′5)
	Effort that can be assumed	31 (38′7)
	Major problems	8 (10)
	Private preferred	6 (7′5)
	Missing	_
Psychological impact	Nothing, forgotten	19 (23′7)
	Moderate	40 (50)
	A before and after	21 (26′3)
	Missing	
Subsequent review	No, nobody informed me	3 (3′8)
	No, she cannot attend	28 (35)
	Yes, with ultrasound	47 (58′7)
	Missing	2 (2′5)
Change of contraceptive method	Yes, hormonal	47 (58′7)
	Yes, non-hormonal	13 (16′3)
	Same, better use	20 (25)
	Missing	_
Aspects to be improved	None	44 (55)
	Speeding up procedures	4 (5)
	Accessibility	11 (13′8)
	Improve treatment	2 (2′5)
	Anaesthesia	3 (3′8)
	Further information	6 (7′5)
	Other	2 (2′5)
	Missing	_

## Data Availability

The datasets used and/or analysed during the current study are available from the corresponding author on reasonable request.

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
