# Peer review of "Reproductive Health Practices in Spanish Women Who Underwent Voluntary Termination of Pregnancy"

_diseases, 2023, doi:10.3390/diseases11010037_

Round 1

Reviewer 1 Report

This paper is well organized and interesting both for science and public health. The incidence of Voluntary Interruption of Pregnancy is an important indicator of unwanted pregnancies. This analysis is essential for monitoring the well-being of women seeking reproductive advice. Socio-demographic data of women who requested Voluntary Interruption of Pregnancy were assessed. A quasi-experimental study was designed without a control group. Socio-demographic and productive health variables were used. A satisfaction survey was carried out. A clear conclusion was that reproductive health education needs to be improved. The women need better accessibility to the intervention and more concrete advice for future reproductive health. These conclusions should be integrated in the public health programs on a general basis. The methodology is robust enough to build public health strategies on this study combined with the general health system information at regional and national level.  

Author Response

Thank you for your comments and feedback.

Reviewer 2 Report

The manuscript covers a very important public health issue related to pregnancy interruption.

The manuscript title sounds clear and concise.

The abstract is properly structured. However, needs improvement. Line 18 - is it necessary to capitalize these words everywhere? "Voluntary Interruption of Pregnancy".

The main text in general is very narrow and does not qualify for a full-length research article. It should be either expanded, or the manuscript should be submitted as a short communication, brief report, or research letter.

Introduction

The introduction part is too narrow and does not give a proper understanding of the study rationale.

1. I would suggest starting with the definition. What do you mean by voluntary pregnancy interruption?

2. Line 18 please define all abbreviations when it is mentioned the first time. Please do it through the whole text. What is VTP, IVE (lines 41, 43).

3. Line 46 - Organic low - please add more details to make it clear for potential international readers. 

4. What are the local leading causes/factors of seeking an artificial abortion?

5. Please provide any available statistics related to the complication after pregnancy termination, including psychological factors. 

6. Why do you want to analyze the socioeconomic characteristics? From the introduction, you give a potential reader, it is completely unclear what drives your aim.

Methods 

This section is very narrow as well.

Please include the following details (you may write separate paragraphs as subheadings): Study design, study subjects (clearly defined inclusion and exclusion criteria), study instruments {survey(s) description}, study variables, ethical considerations, and a description of the statistical analysis. How the sample size was calculated and justified? All these sections are essential parts of the methods section for an original research manuscript. 

Results are clear and supported by good tables and figure.

Discussion 

The discussion part is one of the most important parts of the original research papers, as it presents and discusses the authors' findings. Thus, here it must be restructured to meet the criteria of the research manuscript and to sound scientific. Why do you start this part by presenting your findings? The first sentence of the discussion part sounds weird as it is placed improperly. Restructuring the discussion part is strongly suggested. Please follow the classical structure that a discussion part should consist of:

1.1   Rationale of the study (why it was done)

1.1.1          Main findings of the study

1.1.2          What makes our study unique

1.1.3          What it adds to what we already know

1.2   Study subjects

1.3   Subject of the discussion

Comparison of our results with neighboring countries, with countries of the same development levels  (income), with developed high-income countries). Agreement and disagreement with the studies compared

1.4   Sum up the study, describe the study strengths and limitations

1.5   Clinical implication

2

The references should be updated as most of them are older than 10 years. There are many up-to-date research published more recently.

Author Response

The manuscript covers a very important public health issue related to pregnancy interruption.

The manuscript title sounds clear and concise.

The abstract is properly structured. However, needs improvement. Line 18 - is it necessary to capitalize these words everywhere? "Voluntary Interruption of Pregnancy".

RESPONSE: Done, these words appear in lowercase.

The main text in general is very narrow and does not qualify for a full-length research article. It should be either expanded, or the manuscript should be submitted as a short communication, brief report, or research letter.

Introduction

The introduction part is too narrow and does not give a proper understanding of the study rationale.

RESPONSE: We have added a paragraph and a few sentences to improve the understanding of the justification of the study and ordered the text according to the recommendation.

  1. I would suggest starting with the definition. What do you mean by voluntary pregnancy interruption?

RESPONSE: Done (lines 40-41).

  1. Line 18 please define all abbreviations when it is mentioned the first time. Please do it through the whole text. What is VTP, IVE (lines 41, 43).

RESPONSE: VTP was defined at the start of the introduction.

  1. Line 46 - Organic low - please add more details to make it clear for potential international readers.

RESPONSE: Done (lines 47-49).

  1. What are the local leading causes/factors of seeking an artificial abortion?

RESPONSE: Done (lines 42-44).

  1. Please provide any available statistics related to the complication after pregnancy termination, including psychological factors. 

RESPONSE: Done (lines 44-46).

  1. Why do you want to analyze the socioeconomic characteristics? From the introduction, you give a potential reader, it is completely unclear what drives your aim.

RESPONSE: Done, we have added bibliographic references (1,2,3) to support it.

Methods

This section is very narrow as well.

Please include the following details (you may write separate paragraphs as subheadings): Study design, study subjects (clearly defined inclusion and exclusion criteria), study instruments {survey(s) description}, study variables, ethical considerations, and a description of the statistical analysis. How the sample size was calculated and justified? All these sections are essential parts of the methods section for an original research manuscript.

RESPONSE: Done. We have included all these sections.

Results are clear and supported by good tables and figure.

Discussion

The discussion part is one of the most important parts of the original research papers, as it presents and discusses the authors' findings. Thus, here it must be restructured to meet the criteria of the research manuscript and to sound scientific. Why do you start this part by presenting your findings? The first sentence of the discussion part sounds weird as it is placed improperly. Restructuring the discussion part is strongly suggested. Please follow the classical structure that a discussion part should consist of:

1.1   Rationale of the study (why it was done)

1.1.1          Main findings of the study

1.1.2          What makes our study unique

1.1.3          What it adds to what we already know

1.2   Study subjects

1.3   Subject of the discussion

Comparison of our results with neighboring countries, with countries of the same development levels  (income), with developed high-income countries). Agreement and disagreement with the studies compared

1.4   Sum up the study, describe the study strengths and limitations

1.5   Clinical implication

2 The references should be updated as most of them are older than 10 years. There are many up-to-date research published more recently.

RESPONSE: We have restructured the discussion section following the structure recommended by the reviewer. We have added a first paragraph to justify the interest of this research and a final paragraph on the clinical implication of the study. In addition, we have added more recent bibliographic references (1) 2022, (2) 2016, and (3) 2014.

Reviewer 3 Report

This manuscript tackles an important issue, namely the social correlates of abortion in a major area of Spain. The following revisions would be welcome.

1. Justify the methods in a more compelling fashion. Do power analyses reveal sufficient survey numbers to warrant confidence in the results? I have serious concerns about the relatively small number of women who participated.

2. This is not really a quasi-experimental study as I see it without any form of control group. Please justify the design and describe it in a way that is more consistent with the approach used. This seems like a pilot study that is largely descriptive in nature. 

3. Given the relatively small survey numbers, are there are supplemental forms of data or contextual information that can be used (e.g., health campaign or clinic information shared with women that could be analyzed in some way)?

4. A stronger justification for the site in which this research was conducted would be welcome. Why this locale and this particular time period? What about the religious climate here? Why is this an optimal place in which to conduct this study?

5. More attention to study limitations is warranted, particularly in tandem with promising directions for future research through which these limitations could be addressed. Sample size issues are mentioned, but what about variables left out, lack of qualitative data that could enrich what's known, etc.?

The writing is generally good, but a careful proofreading would improve this manuscript.

Author Response

This manuscript tackles an important issue, namely the social correlates of abortion in a major area of Spain. The following revisions would be welcome.

  1. Justify the methods in a more compelling fashion. Do power analyses reveal sufficient survey numbers to warrant confidence in the results? I have serious concerns about the relatively small number of women who participated.

RESPONSE: We have tried to answer and clarify this question in the section "Methods: Study subjects". In addition, this point is also discussed in the "Strengths and limitations" section.

  1. This is not really a quasi-experimental study as I see it without any form of control group. Please justify the design and describe it in a way that is more consistent with the approach used. This seems like a pilot study that is largely descriptive in nature. 

RESPONSE: We have consistently redrafted the paragraph.

  1. Given the relatively small survey numbers, are there are supplemental forms of data or contextual information that can be used (e.g., health campaign or clinic information shared with women that could be analyzed in some way)?

RESPONSE: National studies on sociodemographic aspects and contraceptive practices of women seeking VTP are old and date back to the previous Organic Law 9/1985; therefore, this research is justified in updating the information on this topic.

  1. A stronger justification for the site in which this research was conducted would be welcome. Why this locale and this particular time period? What about the religious climate here? Why is this an optimal place in which to conduct this study?

RESPONSE: The study was conducted in the health area where the principal investigator worked. This hospital in Salamanca centralizes all abortion requests through the Social Security of that province. The national statistics published on abortions are annual, so we wanted to collect information for at least a full year. The religious environment is not significant. However, demographic factors have been changing as the population has been aging and the birth rate has been declining. (lines 88-91).

  1. More attention to study limitations is warranted, particularly in tandem with promising directions for future research through which these limitations could be addressed. Sample size issues are mentioned, but what about variables left out, lack of qualitative data that could enrich what's known, etc.?

RESPONSE: We have tried to answer this question.

Round 2

Reviewer 2 Report

Thank you very much for considering my comments.

The manuscript has been improved, however, some parts remain unclear.

1.From the introduction part, it is not clear how Voluntary Termination of Pregnancy (VTE) differs from Induced (artificial) abortion.

2. Why did the authors decide to use this terminology, VTE, instead of the widely accepted "abortion", "medical abortion", or "induced abortion"?

3. Lines 53-54 in the introduction - " gestational age limit for terminations due to an extremely serious and incurable disease in the foetus". If there are certain diseases of pregnancy/featus and there is no limit for the termination in terms of the gestational age, then it will be termination of pregnancy due to medical indications (fetal or maternal). And it will be addressed based on the gestational age (abortion or preterm labor). It was defined far ago by WHO.  

 WHO . Recommended definitions, terminology and format for statistical tables related to the perinatal period and use of a new certificate for cause of perinatal deaths. Modifications recommended by FIGO as amended October 14 1976. Acta Obstet Gynecol Scand. 1977;56:247–253

So it is completely unclear from the manuscript why the terminology of VTE is chosen.

This is a cornerstone of the manuscript. Until it is properly clarified or justified, I do not see opportunity to continue the review

Author Response

1.From the introduction part, it is not clear how Voluntary Termination of Pregnancy (VTE) differs from Induced (artificial) abortion.

RESPONSE: Actually, these terms mean the same thing.

  1. Why did the authors decide to use this terminology, VTE, instead of the widely accepted "abortion", "medical abortion", or "induced abortion"?

RESPONSE: The term "abortion" is very general and does not clarify the cause, whether it is spontaneous or voluntary. "Medical abortion" refers to a way of terminating a pregnancy through medication that causes expulsion of the fetus. However, many pregnancy terminations are performed surgically. The term "induced abortion" would be a synonym and would include our cases. However, the term voluntary termination of pregnancy seems to us to be more accurate.

  1. Lines 53-54 in the introduction - " gestational age limit for terminations due to an extremely serious and incurable disease in the foetus". If there are certain diseases of pregnancy/featus and there is no limit for the termination in terms of the gestational age, then it will be termination of pregnancy due to medical indications (fetal or maternal). And it will be addressed based on the gestational age (abortion or preterm labor). It was defined far ago by WHO.  

 WHO . Recommended definitions, terminology and format for statistical tables related to the perinatal period and use of a new certificate for cause of perinatal deaths. Modifications recommended by FIGO as amended October 14 1976. Acta Obstet Gynecol Scand. 1977;56:247–253

So it is completely unclear from the manuscript why the terminology of VTE is chosen.

This is a cornerstone of the manuscript. Until it is properly clarified or justified, I do not see opportunity to continue the review.

RESPONSE: lines 53-61 explain the Spanish law on abortions. Our study is carried out only with the first assumption: women who terminate the pregnancy of their own free will and therefore have to do so before the 14th week. In this case it would be called voluntary interruption of pregnancy or induced abortion.

Reviewer 3 Report

I commend the authors on a capable revision. I continue to believe that the study should not be described as "quasi-experimental" without a comparison group. What makes a study "quasi" is that randomization is missing, but random assignments requires two or more groups. The authors have indicated no randomization, which is correct, because this is a single-group study. The comparison is one group over time, which is not experimental in any fashion. That's OK, as I review (and conduct) studies like this often where control or comparison groups are not always feasible. Can it just be called an "intervention study" with no additional modifier? Please drop any reference to quasi-experimental. 

Author Response

RESPONSE: Done. We agree to define it as “an intervention study" as proposed by the reviewer.